# Towards Better-than-2 Approximation for Constrained Correlation Clustering

Andreas Kalavas [* 1 2]   Evangelos Kipouridis [* 1]   Nithin Varma [* 3]

## Abstract

In the Correlation Clustering problem, we are given an undirected graph and are tasked with computing a clustering (partition of the nodes) that minimizes the sum of the number of edges across different clusters and the number of non-edges within clusters. In the constrained version of this problem, the goal is to compute a clustering that satisfies additional hard constraints mandating certain pairs to be in the same cluster and certain pairs to be in different clusters. Constrained Correlation Clustering is APX-Hard, and the best known approximation factor is 3 (van Zuylen *et al.* [SODA '07]). In this work, we show that in order to obtain a better-than-2 approximation, solving the (exponentially large) Constrained Cluster LP would be sufficient[1].

## 1. Introduction

Clustering is a fundamental unsupervised learning task and has many applications in machine learning and data mining. The high-level goal is to partition a set of elements into *clusters* such that similar elements are in the same cluster and dissimilar elements are in different clusters. Over the years, several definitions of clustering have been considered, each with a different objective.

*Correlation Clustering*, proposed by Bansal, Blum, and Chawla (Bansal et al., 2004), is one such clustering formulation, which was widely studied since its introduction. The appeal of Correlation Clustering stems from its simple and natural definition and from the absence of requirement to specify the number of clusters as part of the input. The problem has found success in many applications including, but not limited to, automated labeling (Agrawal

et al., 2009; Chakrabarti et al., 2008), clustering ensembles (Bonchi et al., 2013), community detection (Chen et al., 2012; Veldt et al., 2018), disambiguation tasks (Kalashnikov et al., 2008), duplicate detection (Arasu et al., 2009) and image segmentation (Kim et al., 2011; Yarkony et al., 2012).

Given a complete unweighted undirected graph $(V, E^+ \uplus E^-)$, whose edges are labeled either "plus" or "minus", the Correlation Clustering problem is to output a partition (*clustering*) $\mathcal{C} = \{C_1, \ldots, C_k\}$ of the vertex set $V$. The sets $C_i$ of $\mathcal{C}$ are called clusters. The objective is to minimize the sum of the number of plus edges across different clusters and the number of minus edges within clusters.

One can view the edges in the input graph as modeling pairwise relationships between entities corresponding to the nodes. Specifically, a plus edge denotes the preference of its endpoints to be clustered together and a minus edge denotes their preference to be separated. The cost of a clustering is then the number of preferences violated by the clustering and the goal of Correlation Clustering translates to finding a clustering that satisfies the preferences of the maximum number of pairs.

The real-world applications that motivate clustering as above often necessitate incorporating additional hard constraints that result from prior knowledge. In such a semi-supervised setting, it is imperative to study the more general problem of constrained clustering. There are several variants of constrained clustering including, but not limited to constrained $k$-means (Wagstaff et al., 2001), spectral clustering with constraints (Wang & Davidson, 2010), constrained ranking and clustering (van Zuylen & Williamson, 2009), and constrained fuzzy clustering (Pham, 2002). Constrained clustering is particularly valuable as it improves the quality of clusters, aligns outcomes with domain-specific knowledge, and reduces sensitivity to noisy data.

In this work, we study Constrained Correlation Clustering, a variant of Correlation Clustering capturing the idea of critical pairs of nodes, which was first studied by van Zuylen and Williamson (van Zuylen & Williamson, 2009). Con-

---

*Equal contribution   [1]Max Planck Institute for Informatics & Saarland Informatics Campus, Saarbrücken, Germany [2]Archimedes, Athena Research Center, Athens, Greece [3]University of Cologne, Germany. Correspondence to: Andreas Kalavas <mathm24008@math.aegean.gr>.

*Proceedings of the 42nd International Conference on Machine Learning*, Vancouver, Canada. PMLR 267, 2025. Copyright 2025 by the author(s).

---

[1]The peer-reviewed version of this article claimed an efficient algorithm for solving the Constrained Cluster LP. An error in the proof, that the authors discovered after the review process, led them to revise the results to be conditional on the existence of a valid LP solution.

strained Correlation Clustering introduces hard constraints in addition to the pairwise preferences, where one could have must-link (requiring nodes to be in the same cluster) or cannot-link (requiring nodes to be separate) constraints. A clustering is valid if it satisfies all hard constraints, and the goal is to find a valid clustering of minimum cost.

Constrained Correlation Clustering with must-link and cannot-link constraints is well-motivated in both theory and practice. For instance, it has been used to cluster news articles about the same event across different languages (Gael & Zhu, 2007). The hard constraints here ensure that news articles about different events from the same language do not end up in the same cluster.

## 1.1. Previous Results

Correlation Clustering was defined by Bansal, Blum, and Chawla (Bansal et al., 2004), who, in addition to proving its NP-Hardness, provided a deterministic constant-factor approximation, the constant being larger than 15,000. Subsequent results improved the approximation guarantee by relying on LP-rounding techniques: Charikar, Guruswami and Wirth showed that the problem is APX-Hard and also gave a deterministic 4-approximation (Charikar et al., 2005), Ailon, Charikar and Newman gave a randomized 2.5-approximation (Ailon et al., 2008), and Chawla, Makarychev, Schramm and Yaroslavtsev gave a deterministic 2.06-approximation (Chawla et al., 2015). The last result is nearly optimal among algorithms that round the natural LP formulation, since its integrality gap is at least 2. In a breakthrough result by Cohen-Addad, Lee and Newman, a $(1.994 + \epsilon)$-approximation using the Sherali-Adams relaxation broke the 2 barrier (Cohen-Addad et al., 2022). The approximation guarantee was later improved to $1.73 + \epsilon$ (Cohen-Addad et al., 2023) by Cohen-Addad, Lee, Li and Newman, and even to $1.437$ by (Cao et al., 2024a). There is also a combinatorial $1.847$-approximation by (Cohen-Addad et al., 2024b) for the problem. The current-state of the art is an adaptation of the $1.437$ approximation that runs in linear (Cao et al., 2025b), and even sublinear (Cao et al., 2025a) time.

Correlation Clustering has also been studied in different settings such as dynamic algorithms (Cohen-Addad et al., 2024a), parameterized algorithms (Fomin et al., 2014), sublinear and streaming algorithms (Assadi & Wang, 2022; Cambus et al., 2024; Behnezhad et al., 2022; 2023; Cambus et al., 2024; Makarychev & Chakrabarty, 2023), massively parallel computation (MPC) algorithms (Cohen-Addad et al., 2021; Cao et al., 2024b), and differentially private algorithms (Bun et al., 2021).

Regarding Constrained Correlation Clustering, the state-of-the-art approximation is given in the work of van Zuylen and Williamson (van Zuylen & Williamson, 2009) who de-

signed a deterministic 3-approximation. We note here that algorithms with better-than-3 approximation for Correlation Clustering existed for a long time after the 3-approximation by (van Zuylen & Williamson, 2009), but they do not seem to extend to the case of Constrained Correlation Clustering. Due to the significance of the problem, a follow-up work by Fischer, Klausen, Kipouridis and Thorup (Fischer et al., 2024) focused on faster algorithms for Constrained Correlation Clustering, even at the expense of worse approximation.

It is worth noting that it is not possible, in general, to handle the must-link constraints by simply merging the nodes in connected components induced by these constraints into supernodes, as such an aggregation results in a weighted instance outside the problem.

## 1.2. Related Work

Closely related to the problems considered by us are hierarchical clustering problems under similar constrained settings. In particular, for each pair $\{u, v\}$ we are given an upper and a lower bound regarding the distance of $u$ and $v$ in the output (Farach, Kannan, and Warnow (Farach et al., 1995)). Ailon and Charikar, in (Ailon & Charikar, 2011), make progress on constrained hierarchical clustering, but optimal algorithms are still elusive. We note that Constrained Correlation Clustering is a special case of constrained hierarchical clustering (where the hierarchy is trivial).

## 1.3. Our Contribution and Techniques

Our main result is a proof that, to achieve a better-than-2 approximation for Constrained Correlation Clustering, it suffices to design an efficient solution to the Constrained Cluster LP (Figure 1).

**Theorem 1.1** (Informal). *If there exists a polynomial-time algorithm to solve the Constrained Cluster LP, then there exists a polynomial time algorithm for Constrained Correlation Clustering whose approximation factor is* $1.92$.

Once again, we note that we do not claim a solution to the Constrained Cluster LP. We consider it plausible because the corresponding unconstrained Cluster LP has been solved in polynomial (Cao et al., 2024a), linear (Cao et al., 2025b), and even sublinear time (Cao et al., 2025a).

There are two main techniques for (unconstrained) Correlation Clustering to achieve a better-than-2 approximation. One (Cao et al., 2024a) is based on the Cluster LP, an LP stronger than the natural one[2], while the other (Cohen-Addad et al., 2024b) is based on a local search technique.

The problem in extending the LP based approach is that, even given a valid solution to the Constrained Cluster LP,

---

[2] In fact it has exponential size, but one can, in polynomial time, compute a $(1 + \varepsilon)$ approximate solution of polynomial size.

its rounding involves creating clusters based on independent sampling of nodes (based on some probability distributions dictated by the LP solution). However, when hard constraints are enforced, we can no longer use independent sampling since, for example, placing a node $u$ in a cluster implies that all nodes with a cannot-link constraint with $u$ cannot be in the same cluster.

The local-search approach, on the other hand, is based on a simple principle: if we have a clustering whose cost is more than 2 times the optimal cost, there exists a cluster in the optimal solution that we can "force" into our clustering[3] and get a clustering with smaller cost. The proof is based on a simple counting argument, and the technique would directly lend itself to a 2 approximation, if we could spend exponential time to find such a cluster. The novelty of (Cohen-Addad et al., 2024b) is that they manage to (nearly) simulate this in polynomial time, and their approximation is $2 + \varepsilon$. As one would expect, achieving this is the most technically challenging part of their paper. It is possible that these techniques could be extended to the Constrained Correlation Clustering setting, but even if true, the resulting algorithm and analysis would be highly non-trivial.

Instead, in this work, we propose a simple approach for Constrained Correlation Clustering that combines the strengths of both aforementioned techniques.

Perhaps the most critical observation in our paper is that even though we do not have access to the optimal clustering, an optimal *fractional* clustering (that is, a solution to the LP) would suffice to guide the local search, and thus bypass the need for the heavy machinery introduced in (Cohen-Addad et al., 2024b). In particular, one could embed the hard constraints in the LP, and guarantee that the output of the local search is a valid clustering.

Despite the success of using the optimal fractional clustering to get a 2-approximation, it does not come without its own shortcomings. In particular, the problem occurs when trying to break the 2 barrier. In (Cohen-Addad et al., 2024b), they show that normally the output of the local search is already a better-than-2 approximation, except for some very specific cases. Then, they run a second local search, which is incentivised to give a very different solution than the first (thus, intuitively, avoiding the specific bad cases). To argue that one of the two solutions achieves a better-than-2 approximation, they show that if both approximations were at least 2, then it would be possible to "mix" 3 different clusterings (the two solutions and the optimal one) in order to obtain a clustering that is better than optimal (a contradiction).

In our case however, it is not even clear what mixing two clusterings and a fractional clustering would mean. A

---

[3]More formally, given a clustering $\{C_1, C_2, \ldots, C_k\}$ and a cluster $C$, the new clustering is $\{C, C_1 \setminus C, C_2 \setminus C, \ldots, C_k \setminus C\}$.

straightforward approach would be to first round the fractional clustering, obtain a clustering $\mathcal{C}$ that is close to optimal, and then mix the two solutions and $\mathcal{C}$. The problem with this approach is that it would normally only show that we can get a better clustering than $\mathcal{C}$, not better than the optimal, which is no longer a contradiction.

Still, we show that there exists one particular way to round the fractional clustering, so that the mixing is guaranteed to be below the optimal, not just $\mathcal{C}$. In fact, this particular rounding simply samples clusters with probabilities proportional to the values of the LP solution. Its properties were analysed already in (Cao et al., 2024a), as one part of their overall algorithm.

Finally, we note that in our case we can algorithmically compute and use $\mathcal{C}$, given access to the fractional optimal (unlike in (Cohen-Addad et al., 2024b) where they do not have access to the optimal clustering, and therefore $\mathcal{C}$ is only part of the analysis).

### 1.4. Organization

The rest of the paper has the following structure. In Section 2, we formally define the problem, and give some basic notation. In Section 3, we describe the LP that models Constrained Correlation Clustering. Finally, in Section 4 we give a full analysis for our main algorithm and how it achieves its approximation factor of $(1.92 + \varepsilon)$.

## 2. Preliminaries

The graphs $(V, E^+ \uplus E^-)$ we consider in this paper are complete, unweighted, undirected, and every edge is labeled plus or minus (it belongs to one of $E^+$ or $E^-$, respectively). We typically set $n = |V|$. Let $\binom{A}{2}$ denote the set of all size-2 subsets of a set $A$. We often abbreviate the (unordered) set $\{u, v\}$ by $uv$. Before introducing our problem, we first define the cost of a clustering:

**Definition 2.1.** Let $\mathcal{E}^+(\mathcal{C})$ and $\mathcal{E}^-(\mathcal{C})$ be the sets of plus and minus edges that are not satisfied in a clustering $\mathcal{C}$. That is $\mathcal{E}^+(\mathcal{C}) = \{uv \in E^+ \mid \nexists C \in \mathcal{C} \text{ with } \{u,v\} \subseteq C\}$ and $\mathcal{E}^-(\mathcal{C}) = \{uv \in E^- \mid \exists C \in \mathcal{C} \text{ with } \{u,v\} \subseteq C\}$. The cost of $\mathcal{C}$ is defined as $\text{cost}(\mathcal{C}) = |\mathcal{E}^+(\mathcal{C})| + |\mathcal{E}^-(\mathcal{C})|$.

We now formally define Constrained Correlation Clustering.

**Definition 2.2** (Constrained Correlation Clustering)**.** Given an instance $(V, E^+ \uplus E^-, F, H)$, where $(V, E^+ \uplus E^-)$ is an edge-labeled graph, $F \subseteq \binom{V}{2}$ is a set of friendly pairs, and $H \subseteq \binom{V}{2}$ is a set of hostile pairs, compute a minimum cost clustering $\mathcal{C} = \{C_1, \ldots, C_k\}$ of $V$ such that no pair $uv \in F$ has $u, v$ in different clusters and no pair $uv \in H$ has $u, v$ in the same cluster.

## 3. LP with constraints

In this section, we describe the linear program used to model the problem. We will use its solution to guide the local searches for the algorithm later.

Given an instance $(V, E^+ \uplus E^-, F, H)$ of Constrained Correlation Clustering, we can define the following linear program, which we call the Constrained Cluster LP. It has two types of variables: $x_{uv}$ for every $uv \in \binom{V}{2}$ and $z_C$ for every $C \subseteq V$. The variable $x_{uv}$ describes the desirability of nodes $u$ and $v$ to be in different clusters (one can view it as a distance between the two nodes) and the variable $z_C$ captures the probability of cluster $C$ to be in the final clustering.

*Figure 1.* Constrained Cluster LP.

$$
\begin{aligned}
\min \quad & \sum_{uv \in E^+} x_{uv} + \sum_{uv \in E^-} (1 - x_{uv}) \\
\text{s.t.} \quad & \sum_{S \ni u} z_S = 1 && \forall u \in V, \\
& \sum_{S \supseteq \{u,v\}} z_S = 1 - x_{uv} && \forall uv \in \binom{V}{2}, \\
& z_S \geq 0 && \forall S \subseteq V, S \neq \emptyset, \\
& x_{uv} = 0 && \forall uv \in F, \\
& x_{uv} = 1 && \forall uv \in H
\end{aligned}
$$

The $\sum_{S \ni u} z_S = 1$ constraints ensure that all vertices belong to some cluster in the final clustering. The $\sum_{S \supseteq \{u,v\}} z_S = 1 - x_{uv}$ constraints state that the closeness of two nodes is proportional to the total weight of clusters containing both of them. Finally, the $x_{uv} = 0$ and $x_{uv} = 1$ constraints model the hard constraints of the instance. We note that without the last two types of constraints, the above LP is identical to the Cluster LP for (unconstrained) Correlation Clustering, solved in (Cao et al., 2024a; 2025b;a).

**Assumption 3.1.** Let OPT be the value of the optimal solution for a Constrained Correlation Clustering instance $(V, E^+ \uplus E^-, F, H)$. We assume that for any constants $c > 0, \varepsilon > 0$ there exists an algorithm that runs in time $n^{\mathsf{poly}(1/\varepsilon)}$ and with probability $1 - n^{-c}$ returns a solution to the Constrained Cluster LP such that:

- its value is at most $(1 + \varepsilon)\mathsf{OPT}$,
- it has at most $\mathsf{poly}(n, \frac{1}{\varepsilon})$ subsets $S$ with $z_S > 0$,
- for every $uv \in F$ there is no subset $S$ with $z_S > 0$ and $|S \cap \{u, v\}| = 1$,
- for every $uv \in H$ there is no subset $S$ with $z_S > 0$ and $S \supseteq \{u, v\}$.

From this point on, we always assume that we have access to a particular solution to the LP, with the properties described in Assumption 3.1.

## 4. Local Search Algorithm

In this section, we describe our algorithm and give a full proof for its approximation factor.

At a high level, the algorithm solves the Constrained Cluster LP (Assumption 3.1) whose solution we view as a (fractional) optimal clustering[4]; we then perform a simple local search procedure (guided by this optimal) that gives a 2-approximation. In fact, this solution already gives a better-than-2 approximation, unless some really specific conditions occur. It turns out that it suffices to run local search once again, but this time we discourage the new clustering from looking similar to the previous one (by adding a penalty in the objective function, when the two clusterings are similar).

In (Cohen-Addad et al., 2024b), they prove that one of the two clusterings gives a better-than-2 approximation as follows: assuming this is not true, they show how to obtain a new clustering that has approximation better than the optimal (a contradiction). To obtain this clustering, they "mix" the optimal with the two local search solutions. We perform the same steps, but in our case we can actually perform these steps as part of the algorithm, not just in the analysis, exactly because in our case we assume we have obtained a (fractional) optimal solution.

### 4.1. Local Search Algorithm

We first define an operation (which we call local move) that transforms a clustering $\mathcal{C}$. Intuitively, the operation takes a new cluster $C$ and swaps it in the clustering; the resulting clustering is defined as $\mathcal{C}_C = \{S \setminus C : S \in \mathcal{C}\} \cup \{C\}$. We call a local move *legal*, if the cluster $C$ that we introduce to the clustering $\mathcal{C}$ does not violate any constraints in $F$ or $H$. In other words, $C$ does not contain both endpoints of a hostile edge nor does it contain only one endpoint of a friendly edge.

**Lemma 4.1.** *Starting from a feasible clustering $\mathcal{C}$, applying any number of legal local moves results in a clustering that does not violate any constraints in $F$ or $H$.*

*Proof.* We consider a single legal local move, as by induction on the number of legal local moves we directly get the lemma.

For a friendly pair to be violated, there must exist a cluster containing exactly one of its endpoints. By the definition of a legal move, this cannot happen in the introduced cluster

---

[4]From this point on, we refer to a solution to the Constrained Cluster LP as a fractional clustering.

$C$ (which either contains both or none of the endpoints). Additionally, this cannot happen in other clusters as well; if it did, then the other endpoint would be in $C$, which we already proved cannot happen. Similarly, for a hostile pair to be violated, the two endpoints must end up in the same cluster. The only cluster that can contain nodes that were in different clusters in $\mathcal{C}$ is cluster $C$, which by definition does not contain both endpoints of a hostile pairs. $\qquad\square$

*Observation* 4.2. We note that it is direct to algorithmically obtain a feasible clustering $\mathcal{C}$, by creating a cluster for each connected component of $(V, F)$. Informally, this gives minimal clusters satisfying $F$. If any hard constraint in $H$ was not satisfied, then the instance itself would be infeasible (two nodes are forced to be together, by $F$, and to be separated, by $H$).

We now define the local search procedure. Let $\mathcal{M}$ be the collection of clusters $C$ with $z_C > 0$ in the LP solution. By Assumption 3.1, there are $\mathrm{poly}(n, \frac{1}{\varepsilon})$ clusters in $\mathcal{M}$. Furthermore, they do not violate the hard constraints, and can thus be used for legal local moves.

---

**Algorithm 1:** Local Search

1   LOCALSEARCH()
2     $\mathcal{L} \leftarrow$ arbitrary feasible clustering
3     **while** $\exists C \in \mathcal{M}$ *such that* $\mathrm{cost}(\mathcal{L}_C) < \mathrm{cost}(\mathcal{L})$ **do**
4       $\mathcal{L} \leftarrow \mathcal{L}_C$
5     **return** $\mathcal{L}$

---

The running time of Algorithm 1 is polynomial, as $|\mathcal{M}| = \mathrm{poly}(n, \frac{1}{\varepsilon})$ and the cost can take only non-negative integer values bounded by $\binom{n}{2}$.

### 4.2. Analysis of Local Search: $2$-approximation

Recall that we view the LP solution as a fractional clustering. Here, we introduce notation to separately capture the cost incurred by plus edges that go across clusters and the cost incurred by minus edges within clusters.

**Definition 4.3.** Let $\mathsf{LP}^+$ and $\mathsf{LP}^-$ denote the costs of the plus and minus edges incurred by the fractional clustering given by the LP solution. That is $\mathsf{LP}^+ = \sum_{uv \in E^+} x_{uv}$ and $\mathsf{LP}^- = \sum_{uv \in E^-} (1 - x_{uv})$.

We slightly abuse notation and use $\mathsf{LP}$ to refer both to the linear program, and the cost of its acquired solution ($\mathsf{LP} = \mathsf{LP}^+ + \mathsf{LP}^-$).

We say a clustering $\mathcal{C}$ is $\alpha$-approximate if $\mathrm{cost}(\mathcal{C}) \leq \alpha \mathsf{LP}$. We first show that $\mathcal{L}$ is 2-approximate. In fact, we prove something even stronger, that we later use to get an algorithm that is better than 2-approximate.

**Lemma 4.4.** *Let $\mathcal{L}$ be as in Algorithm 1. Then,*

$$\mathrm{cost}(\mathcal{L}) \leq 2\mathsf{LP} - \mathsf{LP}^- - \sum_{uv \in \mathcal{E}^+(\mathcal{L}) \cup \mathcal{E}^-(\mathcal{L})} x_{uv},$$

*where $\mathsf{LP}$ denotes the value of the fractional LP solution.*

*Proof.* Let $C$ be a cluster such that $z_C > 0$ in the LP solution returned ($C \in \mathcal{M}$). Recall that, $\mathcal{L}_C$ denotes the clustering obtained by applying a local move with cluster $C$ on a clustering $\mathcal{L}$. This local move can affect an edge (i.e. make it satisfied or unsatisfied) only if the edge has at least one endpoint in cluster $C$ (if not, then the endpoints of the edge do not change cluster, which means that they remain as they are - satisfied or unsatisfied). We say that these edges are covered by $C$. By the local optimality of $\mathcal{L}$, we have $\mathrm{cost}(\mathcal{L}) \leq \mathrm{cost}(\mathcal{L}_C)$. Informally, this means that after swapping cluster $C$ in, more edges covered by $C$ are unsatisfied than before. Let $C^+$ denote the set of plus-edges that have exactly one endpoint in $C$. Similarly, let $C^-$ denote the set of minus edges that have both endpoints in $C$. The set $C^+ \cup C^-$ is the set of edges unsatisfied by the cluster $C$. Next, let $\mathcal{E}_C^+(\mathcal{L}) \subseteq \mathcal{E}^+(\mathcal{L})$ be the set of plus edges $uv \in \mathcal{E}^+(\mathcal{L})$ that are covered by $C$, i.e., $|\{u, v\} \cap C| \geq 1$. Let $\mathcal{E}_C^-(\mathcal{L}) \subseteq \mathcal{E}^-(\mathcal{L})$ be the set of minus edges $uv \in \mathcal{E}^-(\mathcal{L})$ such that $|\{u, v\} \cap C| \geq 1$. From the above discussion, for each cluster $C$ with $z_C > 0$, we have, $|C^+| + |C^-| \geq |\mathcal{E}_C^+(\mathcal{L})| + |\mathcal{E}_C^-(\mathcal{L})|$. This further implies that

$$\sum_C z_C \cdot (|C^+| + |C^-|) \geq \sum_C z_C \cdot (|\mathcal{E}_C^+(\mathcal{L})| + |\mathcal{E}_C^-(\mathcal{L})|).$$

We simplify the left-hand side of the above inequality by separately accounting for the contributions of plus and minus edges to the summation. The contribution of a plus-edge $uv$ to the left-hand side of the above sum is simply $\sum_{C:|\{u,v\} \cap C|=1} z_C$. Recall, from the LP in Figure 1, that $\sum_{C:u \in C} z_C = \sum_{C:v \in C} z_C = 1$ and $\sum_{C:\{u,v\} \subseteq C} z_C = 1 - x_{uv}$. From this, it follows that $\sum_{C:|\{u,v\} \cap C|=1} z_C = 2x_{uv}$. Similarly, one can see that the contribution of a minus-edge $uv$ to the left-hand side of the sum is $1 - x_{uv}$. Summing over the plus and minus edges separately,

$$\sum_C z_C \cdot (|C^+| + |C^-|) = 2\mathsf{LP}^+ + \mathsf{LP}^- = 2\mathsf{LP} - \mathsf{LP}^-.$$

We similarly transform the right-hand side $\sum_C z_C \cdot (|\mathcal{E}_C^+(\mathcal{L})| + |\mathcal{E}_C^-(\mathcal{L})|)$ of the above inequality. Each plus edge $uv$ that contributes to $\sum_C z_C \cdot (|\mathcal{E}_C^+(\mathcal{L})| + |\mathcal{E}_C^-(\mathcal{L})|)$ belongs to $\mathcal{E}^+(\mathcal{L})$. A plus edge in $\mathcal{E}^+(\mathcal{L})$ contributes an amount of $z_C$ to the sum if either $u$ or $v$ is contained in $C$. Thus, its total contribution is $\sum_{C:|C \cap \{u,v\}| \geq 1} z_C$, which is equal to $1 + x_{uv}$. Using a similar argument, the contribution

of a minus edge $uv$ in $\mathcal{E}^-(\mathcal{L})$ to the sum can also be seen to be $1 + x_{uv}$. Thus, the summation simplifies to

$$\sum_C z_C \cdot (|\mathcal{E}_C^+(\mathcal{L})| + |\mathcal{E}_C^-(\mathcal{L})|) = \mathsf{cost}(\mathcal{L}) + \sum_{uv \in \mathcal{E}^+(\mathcal{L}) \cup \mathcal{E}^-(\mathcal{L})} x_{uv}.$$

Substituting these rewritten terms in the inequality, we get the desired statement. $\qquad\square$

The last lemma directly gives an approximation ratio of 2, which we reduce further on.

**Corollary 4.5.** *Let $\mathcal{L}$ be as in Algorithm 1. Then,*

$$\mathsf{cost}(\mathcal{L}) \leq 2\mathsf{LP} - \mathsf{LP}^- - \sum_{uv \in E^+(\mathcal{L})} x_{uv}$$

$$\mathsf{cost}(\mathcal{L}) \leq 2\mathsf{LP} - |\mathcal{E}^-(\mathcal{L})| - \sum_{uv \in E^+(\mathcal{L})} x_{uv}$$

*Proof.* The first inequality is just Lemma 4.4 with the term $-\sum_{uv \in \mathcal{E}^-(\mathcal{L})} x_{uv}$ dropped. The second inequality starts from Lemma 4.4 and uses the fact that:

$$\mathsf{LP}^- + \sum_{uv \in \mathcal{E}^-(\mathcal{L})} x_{uv} \geq \sum_{uv \in \mathcal{E}^-(\mathcal{L})} (1 - x_{uv} + x_{uv})$$

where the right-hand side is equal to $|\mathcal{E}^-(\mathcal{L})|$. $\qquad\square$

### 4.3. Local Search with Penalties

Let $\delta = \frac{2}{25}$. For the rest of the proof, it is instructive to think about $\delta$ as a very small constant.

Directly from Corollary 4.5, we observe that either $\mathcal{L}$ is already better than 2-approximate, or some very particular conditions happen:

- Regarding minus-edges, the cost of both $\mathcal{L}$ and the LP is negligible.

- Regarding plus-edges paid by $\mathcal{L}$, the corresponding cost of the LP is negligible.

We formalize this intuition in the following corollary.

**Corollary 4.6.** *If $\mathcal{L}$ is not $(2-\delta)$-approximate, then:*

$$\mathsf{LP}^- + \sum_{uv \in \mathcal{E}^+(\mathcal{L})} x_{uv} < \delta\mathsf{LP},$$

$$|\mathcal{E}^-(\mathcal{L})| + \sum_{uv \in \mathcal{E}^+(\mathcal{L})} x_{uv} < \delta\mathsf{LP}$$

*Proof.* Immediately from Corollary 4.5. $\qquad\square$

Following from the above, if $\mathcal{L}$ is not $(2 - \delta)$-approximate, then its cost is dominated by the plus edges. The idea here is that we try to run a local search that produces a clustering that looks very different from $\mathcal{L}$ (and thus ideally does not satisfy the aforementioned conditions). To do that, we simply penalize the plus edges paid by $\mathcal{L}$ in the cost function (Algorithm 2). We note that this part is exactly as in (Cohen-Addad et al., 2024b).

**Definition 4.7.** For a clustering $\mathcal{C}$ the new cost function is:

$$\mathsf{cost}_{\mathcal{L}}(\mathcal{C}) = \mathsf{cost}(\mathcal{C}) + |\mathcal{E}^+(\mathcal{C}) \cap \mathcal{E}^+(\mathcal{L})|$$

---

**Algorithm 2:** Local Search with penalties

1 $\mathsf{cost}_{\mathcal{L}}(\mathcal{L}')$
2 $\quad$ **return** $\mathsf{cost}(\mathcal{L}') + |\mathcal{E}^+(\mathcal{L}') \cap \mathcal{E}^+(\mathcal{L})|$
3 LOCALSEARCH-WITH-PENALTY()
4 $\quad$ $\mathcal{L} \leftarrow$ LOCALSEARCH()
5 $\quad$ $\mathcal{L}' \leftarrow$ arbitrary feasible clustering
6 $\quad$ **while** $\exists C \in \mathcal{M}$ *such that* $\mathsf{cost}_{\mathcal{L}}(\mathcal{L}'_C) < \mathsf{cost}_{\mathcal{L}}(\mathcal{L}')$
$\quad\quad$ **do**
7 $\quad\quad$ $\lfloor$ $\mathcal{L}' \leftarrow \mathcal{L}'_C$
8 $\quad$ **return** $\mathcal{L}'$

---

Working exactly as for Corollary 4.6, but using $\mathsf{cost}_{\mathcal{L}}$ instead of $\mathsf{cost}$, we get similar conditions for when $\mathcal{L}'$ is not better than 2-approximate:

**Lemma 4.8.** *If $\mathcal{L}'$ is not $(2 - \delta)$-approximate, then:*

$$|\mathcal{E}^-(\mathcal{L}')| + \sum_{uv \in \mathcal{E}^+(\mathcal{L}')} x_{uv} + |\mathcal{E}^+(\mathcal{L}') \cap \mathcal{E}^+(\mathcal{L})|$$

$$< \delta\mathsf{LP} + 2 \sum_{uv \in \mathcal{E}^+(\mathcal{L})} x_{uv}$$

*Proof.* Proved in Appendix A. $\qquad\square$

Directly by Corollary 4.6 and Lemma 4.8 we note that if neither $\mathcal{L}$ nor $\mathcal{L}'$ are $(2-\delta)$-approximate, then the following conditions occur:

- Regarding minus edges, the cost of LP, the cost of $\mathcal{L}$, and the cost of $\mathcal{L}'$ are all negligible.

- Regarding plus edges, not too many of them can contribute to both $\mathcal{L}$ and $\mathcal{L}'$, and, moreover, the ones that contribute to $\mathcal{L}$ or $\mathcal{L}'$ cannot contribute much to the cost of the LP.

The following corollary formalizes the above.

**Corollary 4.9.** *If neither $\mathcal{L}$ and $\mathcal{L}'$ are $(2 - \delta)$-approximate, then:*

$$\mathsf{LP}^- + |\mathcal{E}^-(\mathcal{L})| + |\mathcal{E}^-(\mathcal{L}')|$$
$$+ \sum_{uv \in \mathcal{E}^+(\mathcal{L})} x_{uv} + \sum_{uv \in \mathcal{E}^+(\mathcal{L}')} x_{uv} + |\mathcal{E}^+(\mathcal{L}) \cap \mathcal{E}^+(\mathcal{L}')|$$
$$< 4\delta\mathsf{LP}.$$

*Proof.* Follows from the inequalities of Corollary 4.6 and Lemma 4.8. $\qquad\square$

### 4.4. Final Algorithm and Analysis

The structure of the rest of the proof in (Cohen-Addad et al., 2024b) is as follows: assuming that none among $\mathcal{L}, \mathcal{L}'$ are $(2 - \delta)$-approximate, the authors construct a clustering that is better than 1-approximate (a contradiction). To construct this clustering, they propose a way to "mix" (in the analysis) $\mathcal{L}, \mathcal{L}'$, and the optimal clustering. For this to work in our case, we are required to use the fractional optimal clustering. This is so because the guarantees for our two clusterings $\mathcal{L}, \mathcal{L}'$ are with respect to the fractional optimal instead of the actual optimal, as was done in the paper that introduced the local search approach for Correlation Clustering (Cohen-Addad et al., 2024b). However, the mixing procedure cannot be straightforwardly extended to fractional clusterings. Instead, we first round the fractional clustering $\mathcal{C}^*$ to an integral one, whose properties are crucially very close to that of $\mathcal{C}^*$.

This rounding (see (Cao et al., 2024a)) gives a 2-approximate clustering that samples clusters from $\mathcal{M}$.

---

**Algorithm 3:** Simple sampling-based 2-approximation

---

1   SAMPLING()
2    $\mathcal{I} \leftarrow \emptyset, S \leftarrow V$
3    **while** $S \neq \emptyset$ **do**
4      Sample a cluster $C$ with probability proportional to $z_C$
5      $\mathcal{I} \leftarrow \mathcal{I} \cup \{C \cap S\}$
6      $S \leftarrow S \setminus C$
7    **return** $\mathcal{I}$

---

Algorithm 3 can trivially run in polynomial time by avoiding sampling clusters $C$ with $C \cap S = \emptyset$. Proving that it is a 2-approximation follows directly from Lemma 4.10.

**Lemma 4.10** (Generalization of Lemma 6 of (Cao et al., 2024a)). *For every $uv \in \binom{V}{2}$ the probability that $u, v$ are separated in $\mathcal{I}$ returned by Algorithm 3 is $\frac{2x_{uv}}{1+x_{uv}}$.*

*Proof.* Directly from the algorithm, the probability that $u, v$ are separated is equal to the probability that we select a cluster containing only one of them, conditioned on the fact that we select a cluster containing at least one of them. This is $\frac{\sum_{|S \cap \{u,v\}|=1} z_S}{\sum_{|S \cap \{u,v\}|\geq 1} z_S} = \frac{\sum_{S \ni u, S \not\ni v} z_S + \sum_{S \ni v, S \not\ni u} z_S}{\sum_{S \ni v} z_S + \sum_{S \ni u, S \not\ni v} z_S}$. By the LP in Figure 1 we have $\sum_{S \ni v} z_S = 1$ and $\sum_{S \ni u, S \not\ni v} z_S = \sum_{S \ni u} z_S - \sum_{S \supseteq \{u,v\}} z_S = 1 - (1 - x_{uv}) = x_{uv}$, which proves the claim. $\qquad\square$

**Corollary 4.11.** *It holds that*

$$\mathbb{E}\big[|\mathcal{E}^-(\mathcal{I})| + |\mathcal{E}^+(\mathcal{I}) \cap \mathcal{E}^+(\mathcal{L})| + |\mathcal{E}^+(\mathcal{I}) \cap \mathcal{E}^+(\mathcal{L}')|\big] \leq$$
$$\mathsf{LP}^- + 2 \cdot \sum_{uv \in \mathcal{E}^+(\mathcal{L})} x_{uv} + 2 \cdot \sum_{uv \in \mathcal{E}^+(\mathcal{L}')} x_{uv}$$

*where the expectation is over the randomness of Algorithm 3.*

*Proof.* Notice that for a minus edge $uv$, LP pays $1 - x_{uv}$, while the expected cost of $\mathcal{I}$ is $1 - \frac{2x_{uv}}{1+x_{uv}} = \frac{1}{1+x_{uv}}(1 - x_{uv}) \leq 1 - x_{uv}$. Similarly, for the other two inequalities: for a plus edge $uv$ the cost of LP solution is $x_{uv}$ while the expected cost of $\mathcal{I}$ is $\frac{2x_{uv}}{1+x_{uv}} \leq 2x_{uv}$. $\qquad\square$

**Corollary 4.12.** *For any constants $c > 0, \varepsilon > 0$, we can run Algorithm 3 for $O(\log n)$ times, to get a clustering $\mathcal{I}$ such that*

$$|\mathcal{E}^-(\mathcal{I})| + |\mathcal{E}^+(\mathcal{I}) \cap \mathcal{E}^+(\mathcal{L})| + |\mathcal{E}^+(\mathcal{I}) \cap \mathcal{E}^+(\mathcal{L}')| \leq$$
$$(1 + \varepsilon)(\mathsf{LP}^- + 2 \cdot \sum_{uv \in \mathcal{E}^+(\mathcal{L})} x_{uv} + 2 \cdot \sum_{uv \in \mathcal{E}^+(\mathcal{L}')} x_{uv})$$

*with probability at least $1 - n^{-c}$.*

*Proof.* By Markov Inequality and Corollary 4.11 we have probability at most $1/(1 + \varepsilon) = 1 - \frac{\varepsilon}{1+\varepsilon} < e^{-\frac{\varepsilon}{1+\varepsilon}}$ to get a clustering $\mathcal{I}$ that does not satisfy the claim on a particular execution of Algorithm 3. Therefore the probability of failure in $\frac{1+\varepsilon}{\varepsilon} c \ln n$ repetitions is $\leq e^{-c \ln n} = n^{-c}$. $\qquad\square$

Finally, we show that $\mathcal{I}$ is a clustering satisfying the hard constraints.

**Lemma 4.13.** *Clustering $\mathcal{I}$ returned by Algorithm 3 satisfies all hard constraints.*

*Proof.* By Assumption 3.1, if an endpoint of a friendly pair $uv \in F$ is contained in a cluster $C$ with $z_C > 0$, then the other endpoint will also be contained in that cluster. Thus, a sampled cluster will always have both endpoints of a friendly pair. This means that endpoints of friendly pairs are removed from $S$ simultaneously. Therefore, when introducing $C \cap S$, it either contains both $u$ and $v$ or neither of them. Analogously, by Assumption 3.1 endpoints of hostile pairs are never in the same cluster $C$ with $z_C > 0$. Hence, they cannot end up in the same cluster in clustering $\mathcal{I}$. $\qquad\square$

We are now ready to show that if Algorithms 1 and 2 return clusterings that are not $(2 - \delta)$-approximate, then we can create a new clustering that is $24\delta$-approximate, which is a better-than-2 approximate clustering for our setting of $\delta$.

To do that, we first observe that if neither $\mathcal{L}$ nor $\mathcal{L}'$ are $(2 - \delta)$-approximate, then the following conditions occur:

- Regarding minus edges, the cost of $\mathcal{I}$, the cost of $\mathcal{L}$, and the cost of $\mathcal{L}'$ are all negligible.

- Regarding plus edges, most of them only contribute to the cost of at most one among $\mathcal{L}, \mathcal{L}', \mathcal{I}$.

This is similar to Corollary 4.9, and uses the fact that by Corollary 4.12 clustering $\mathcal{I}$ behaves very similar to the LP.

**Corollary 4.14.** *For any constants $c > 0, \varepsilon > 0$, if neither of $\mathcal{L}$ and $\mathcal{L}'$ are $(2 - \delta)$-approximate, and we obtain $\mathcal{I}$ by using Corollary 4.12 then with probability at least $1 - n^{-c}$:*

$$|\mathcal{E}^-(\mathcal{I})| + |\mathcal{E}^-(\mathcal{L})| + |\mathcal{E}^-(\mathcal{L}')| +$$
$$|\mathcal{E}^+(\mathcal{I}) \cap \mathcal{E}^+(\mathcal{L})| + |\mathcal{E}^+(\mathcal{I}) \cap \mathcal{E}^+(\mathcal{L}')| + |\mathcal{E}^+(\mathcal{L}) \cap \mathcal{E}^+(\mathcal{L}')|$$
$$< (8 + \varepsilon)\delta\mathsf{LP}$$

*Proof.* Beginning from the left hand side and using Corollary 4.12 we get that with probability at least $1 - n^{-c}$:

$$|\mathcal{E}^-(\mathcal{I})| + |\mathcal{E}^-(\mathcal{L})| + |\mathcal{E}^-(\mathcal{L}')| +$$
$$|\mathcal{E}^+(\mathcal{I}) \cap \mathcal{E}^+(\mathcal{L})| + |\mathcal{E}^+(\mathcal{I}) \cap \mathcal{E}^+(\mathcal{L}')| + |\mathcal{E}^+(\mathcal{L}) \cap \mathcal{E}^+(\mathcal{L}')|$$
$$\leq |\mathcal{E}^-(\mathcal{L})| + |\mathcal{E}^-(\mathcal{L}')| + |\mathcal{E}^+(\mathcal{L}) \cap \mathcal{E}^+(\mathcal{L}')| +$$
$$(1 + \varepsilon)(\mathsf{LP}^- + 2 \cdot \sum_{uv \in \mathcal{E}^+(\mathcal{L})} x_{uv} + 2 \cdot \sum_{uv \in \mathcal{E}^+(\mathcal{L}')} x_{uv})$$

Now, we further manipulate the right hand side, by factoring out the constants:

$$|\mathcal{E}^-(\mathcal{L})| + |\mathcal{E}^-(\mathcal{L}')| + |\mathcal{E}^+(\mathcal{L}) \cap \mathcal{E}^+(\mathcal{L}')| +$$
$$(1 + \varepsilon)(\mathsf{LP}^- + 2 \cdot \sum_{uv \in \mathcal{E}^+(\mathcal{L})} x_{uv} + 2 \cdot \sum_{uv \in \mathcal{E}^+(\mathcal{L}')} x_{uv})$$
$$\leq 2(1 + \varepsilon)(\mathsf{LP}^- + \sum_{uv \in \mathcal{E}^+(\mathcal{L})} x_{uv} + \sum_{uv \in \mathcal{E}^+(\mathcal{L}')} x_{uv} +$$
$$|\mathcal{E}^-(\mathcal{L})| + |\mathcal{E}^-(\mathcal{L}')| + |\mathcal{E}^+(\mathcal{L}) \cap \mathcal{E}^+(\mathcal{L}')|)$$

By Corollary 4.9 this is less than $(8 + 8\varepsilon)\delta\mathsf{LP}$, which proves the claim by scaling $\varepsilon$. □

We will now use clusterings $\mathcal{L}, \mathcal{L}'$ and $\mathcal{I}$ to create a new clustering $\mathcal{P}$ such that:

- If $\mathcal{P}$ pays for a minus edge, then at least one of $\mathcal{L}, \mathcal{L}'$ and $\mathcal{I}$ pays for this edge.

- If $\mathcal{P}$ pays for a plus edge, then at least two of $\mathcal{L}, \mathcal{L}'$ and $\mathcal{I}$ pays for this edge.

Therefore, by Corollary 4.14, the total cost of $\mathcal{P}$ is small.

We now describe how to construct $\mathcal{P}$ (Algorithm 4). First, we assign to each node $u$ a label $(X, Y, Z)$, such that $u$ belongs to clusters $X \in \mathcal{L}, Y \in \mathcal{L}'$, and $Z \in \mathcal{I}$.

**Definition 4.15.** An equivalence class $Q_{XYZ}$ is the set of all nodes that have label equal to $(X, Y, Z)$.

The construction of the new clustering $\mathcal{P}$ is shown in Algorithm 4:

---

**Algorithm 4:** Pivoting Procedure

---

1   $\text{PIVOT}(\mathcal{L}, \mathcal{L}')$

2     Obtain $\mathcal{I}$ by Corollary 4.12

3     Assign labels $(X, Y, Z)$ to nodes $u \in V$

4     $V' \leftarrow V, \mathcal{P} \leftarrow \emptyset$

5     **while** $V' \neq \emptyset$ **do**

6       $S \leftarrow$ an arbitrary equivalence class $Q_{XYZ}$ with all its nodes in $V'$ and maximum cardinality

7       $C \leftarrow S \cup \{u \in V' : \text{the label of } u \text{ differs in exactly one label variable from } (X, Y, Z)\}$

8       $\mathcal{P} \leftarrow \mathcal{P} \cup \{C\}, V' \leftarrow V' \setminus C$

9     **return** $\mathcal{P}$

---

**Lemma 4.16.** *For any constants $c > 0, \varepsilon > 0$, if neither $\mathcal{L}$ and $\mathcal{L}'$ are $(2 - \delta)$-approximate, then $\mathcal{P}$ is $(24 + 3\varepsilon)\delta$-approximate with probability at least $1 - n^{-c}$.*

*Proof.* Notice that directly from the algorithm, if two nodes are in the same equivalence class, then they end up in the same cluster of $\mathcal{P}$; therefore $\mathcal{P}$ does not pay for plus edges with endpoints in the same equivalence class (that is, endpoints that do not differ in any label variable). Furthermore, if $u, v$ are in the same cluster in $\mathcal{P}$, it is because they share at least two label variables with some equivalence class $Q_{XYZ}$. Therefore, by pigeonhole principle, $u, v$ share at least one label variable, meaning that $\mathcal{P}$ does not pay for minus edges with their endpoints differing in all label variables.

We conclude that there are 3 categories of edges that we contribute towards the cost of $\mathcal{P}$:

1. minus edges whose endpoints differ in at most two label variables.

2. plus edges whose endpoints differ in exactly one label variable.

3. plus edges whose endpoints differ in at least two label variables.

In Case 1, minus edges are bounded by Corollary 4.14, as at least one of the clusterings $\mathcal{L}$, $\mathcal{L}'$ and $\mathcal{I}$ pay for them (at least one variable in the labels of their endpoints is the same, meaning that there is a cluster in one of $\mathcal{L}$, $\mathcal{L}'$ or $\mathcal{I}$ that contains both). Similarly, in Case 3, plus edges are also bounded by Corollary 4.14, as at least two of the clusterings $\mathcal{L}$, $\mathcal{L}'$ and $\mathcal{I}$ pay for them (at least two variables in the labels of their endpoints are different, therefore they are in different clusters in at least two of the clusterings).

For every edge $uv$ in Case 2, we show that it can be charged to some edge of the previous cases, and each edge from the previous cases is charged by at most two edges of Case 2. This results in cost $3 \cdot (8 + \varepsilon)\delta\mathsf{LP}$, by Corollary 4.14.

Consider an iteration of Algorithm 4 that creates a cluster $C$ in $\mathcal{P}$ by selecting an equivalence class $S = Q_{XYZ}$. For a plus edge $uv$ in Case 2 that we pay for, it must be that one of its endpoints differs in one label variable from $(X, Y, Z)$ (therefore it is in $C$), and the other differs in two (so that it is not in $C$, because we pay for $uv$, but still only differs in one label variable from $u$). W.l.o.g. assume $v \in Q_{XY'Z'}$ and $u \in Q_{XY'Z}$ or $u \in Q_{XYZ'}$, meaning that $v$ can be associated with at most $|Q_{XY'Z}| + |Q_{XYZ'}|$ edges in Case 2 that we pay for.

By maximality of $S$ we have $Q_{XY'Z} \leq S$, and therefore we can find an injective function $f : Q_{XY'Z} \to S$. If $\{v, f(u)\}$ is a plus edge, then it is a Case 3 edge and if $\{v, f(u)\}$ is a minus edge, then it is a Case 1 edge. Similarly for $u \in Q_{XYZ'}$, which proves our claim. $\square$

**Lemma 4.17.** *$\mathcal{P}$ does not violate any hard constraints.*

*Proof.* A friendly pair $uv$ will be in the same cluster in all 3 clusterings $\mathcal{L}$, $\mathcal{L}'$, $\mathcal{I}$ (as they are feasible clusterings), thus $u$ and $v$ will be in the same equivalence class $Q_{XYZ}$. By the construction of $\mathcal{P}$, all nodes of an equivalence class end up in the same cluster. Similarly, a hostile pair $uv$ will have its endpoints $u$ and $v$ in different clusters in all 3 clusterings $\mathcal{L}$, $\mathcal{L}'$, $\mathcal{I}$, which means that their labels will differ in all three variables. Again, by construction, no two nodes that differ in all the label variables end up in the same group in $\mathcal{P}$. $\square$

Our final algorithm returns the best out of all clusterings.

**Theorem 4.18.** *Under Assumption 3.1, for any constants $c > 0, \varepsilon > 0$, there exists a polynomial time algorithm for Constrained Correlation Clustering whose approximation factor is $(1.92 + \varepsilon)$ with probability $1 - n^{-c}$.*

*Proof.* $\mathcal{L}, \mathcal{L}'$ and $\mathcal{P}$ all satisfy the hard constraints (Lemmas 4.1, 4.17) and can be computed in polynomial time.

---

**Algorithm 5:** Main Algorithm

1   $\mathcal{L} \leftarrow \text{LocalSearch}()$
2   $\mathcal{L}' \leftarrow \text{LocalSearch-With-Penalty}()$
3   $\mathcal{P} \leftarrow \text{Pivot}(\mathcal{L}, \mathcal{L}')$
4   $\mathcal{C} \leftarrow \arg\min_{\mathcal{S} \in \{\mathcal{L}, \mathcal{L}', \mathcal{P}\}} \mathsf{cost}(\mathcal{S})$
5   **return** $\mathcal{C}$

---

Furthermore, with probability $1 - n^{-c}$ we successfully solve the LP in Figure 1 (Assumption 3.1) and get that either $\min\{\mathsf{cost}(\mathcal{L}), \mathsf{cost}(\mathcal{L}')\} \leq 1.92\mathsf{LP}$ or $\mathsf{cost}(\mathcal{P}) \leq (1.92 + \varepsilon)\mathsf{LP}$ (by Lemma 4.16 and $\delta = \frac{2}{25}$). As $\mathsf{LP} \leq (1 + \varepsilon)\mathsf{OPT}$ (Assumption 3.1), Algorithm 5 satisfies the claim by scaling $\varepsilon$. $\square$

## 5. Concluding Remarks

In this paper, we present a better-than-2 approximation algorithm for Constrained Correlation Clustering, conditioned on an efficient solution to the Constrained Cluster LP. We obtain our algorithm by combining the assumed Cluster LP solution with a local search technique. Our algorithm is conceptually simpler to analyze and gives a better-than-2 approximation for the special case of Correlation Clustering.

The main open direction is to determine whether there exists an efficient solution for the Constrained Cluster LP. It would also be interesting to explore whether our approach of combining the LP and local search has broader applications. Another open direction is with respect to the inapproximability of Constrained Correlation Clustering. Finally, it is intriguing to know whether inapproximability results stronger than the ones directly implied by those for Correlation Clustering exist for the constrained setting.

**Acknowledgments.** We thank Vincent Cohen-Addad and David Rasmussen Lolck for helpful discussions. We also thank the anonymous reviewers for their feedback that significantly improved the presentation of our paper.

## Impact Statement

This paper presents work whose goal is to advance the field of Machine Learning. There are many potential societal consequences of our work, none which we feel must be specifically highlighted here.

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

## A. Analysis of Local Search with Penalty

In this section, we present the analysis of Algorithm 2, which very closely follows the analysis of Algorithm 1.

**Lemma 4.8.** *If $\mathcal{L}'$ is not $(2 - \delta)$-approximate, then:*

$$|\mathcal{E}^-(\mathcal{L}')| + \sum_{uv \in \mathcal{E}^+(\mathcal{L}')} x_{uv} + |\mathcal{E}^+(\mathcal{L}') \cap \mathcal{E}^+(\mathcal{L})|$$

$$< \delta\mathsf{LP} + 2 \sum_{uv \in \mathcal{E}^+(\mathcal{L})} x_{uv}$$

*Proof.* Let $C$ be a cluster such that $z_C > 0$ in the LP solution returned ($C \in \mathcal{M}$). Recall that, $\mathcal{L}'_C$ denotes the clustering obtained by applying a local move with cluster $C$ on a clustering $\mathcal{L}'$. This local move can affect an edge (i.e. make it satisfied or unsatisfied) only if the edge has at least one endpoint in cluster $C$ (if not, then the endpoints of the edge do not change cluster, which means that they remain as they are - satisfied or unsatisfied). We say that these edges are covered by $C$. By the local optimality of $\mathcal{L}'$, we have $\mathrm{cost}'(\mathcal{L}') \leq \mathrm{cost}'(\mathcal{L}'_C)$. This means that after swapping cluster $C$ in, more edges covered by $C$ are unsatisfied than before. Let $C^+$ denote the set of plus-edges that have exactly one endpoint in $C$. Similarly, let $C^-$ denote the set of minus edges that have both endpoints in $C$. The set $C^+ \cup C^-$ is the set of edges unsatisfied by the cluster $C$. Next, let $\mathcal{E}^+_C(\mathcal{L}') \subseteq \mathcal{E}^+(\mathcal{L}')$ be the set of plus edges $uv \in \mathcal{E}^+(\mathcal{L}')$ that are covered by $C$, i.e., $|\{u, v\} \cap C| \geq 1$. Let $\mathcal{E}^-_C(\mathcal{L}') \subseteq \mathcal{E}^-(\mathcal{L}')$ be the set of minus edges $uv \in \mathcal{E}^-(\mathcal{L}')$ such that $|\{u, v\} \cap C| \geq 1$. From the above discussion, for each cluster $C$ with $z_C > 0$, we have, $|C^+| + |C^+ \cap \mathcal{E}^+(\mathcal{L})| + |C^-| \geq |\mathcal{E}^+_C(\mathcal{L}')| + |\mathcal{E}^+_C(\mathcal{L}') \cap \mathcal{E}^+(\mathcal{L})| + |\mathcal{E}^-_C(\mathcal{L}')|$. This further implies that

$$\sum_C z_C \cdot (|C^+| + |C^-|) + \sum_C z_C \cdot |C^+ \cap \mathcal{E}^+(\mathcal{L})| \geq \sum_C z_C \cdot (|\mathcal{E}^+_C(\mathcal{L}')| + |\mathcal{E}^-_C(\mathcal{L}')|) + \sum_C z_C \cdot |\mathcal{E}^+_C(\mathcal{L}') \cap \mathcal{E}^+(\mathcal{L})|$$

We simplify the first term of every side of the inequality in the same manner as in Lemma 4.4, which gives us:

$$2\mathsf{LP} - \mathsf{LP}^- + \sum_C z_C \cdot |C^+ \cap \mathcal{E}^+(\mathcal{L})| \geq \mathrm{cost}(\mathcal{L}') + \sum_{uv \in \mathcal{E}^+(\mathcal{L}') \cup \mathcal{E}^-(\mathcal{L}')} x_{uv} + \sum_C z_C \cdot |\mathcal{E}^+_C(\mathcal{L}') \cap \mathcal{E}^+(\mathcal{L})|$$

We simplify the term $\sum_C z_C \cdot |C^+ \cap \mathcal{E}^+(\mathcal{L})|$ of the left-hand side by accounting the contributions of plus edges to the summation.

A plus edge in $\mathcal{E}^+(\mathcal{L})$ contributes an amount of $z_C$ to the sum if either $u$ or $v$ is contained in $C$. Thus, its total contribution is $\sum_{C:|C \cap \{u,v\}| \geq 1} z_C$, which is equal to $1 + x_{uv}$.

For an edge to contribute to the term it must be in $\mathcal{E}^+(\mathcal{L})$ and only have one of its endpoints in cluster $C$. This means that every edge in $\mathcal{E}^+(\mathcal{L})$ contributes exactly $\sum_{C:|\{u,v\} \cap C| = 1} z_C = 2x_{uv}$. Thus, we get:

$$\sum_C z_C \cdot |C^+ \cap \mathcal{E}^+(\mathcal{L})| = 2 \sum_{uv \in \mathcal{E}^+(\mathcal{L})} x_{uv}$$

We similarly transform the term $\sum_C z_C \cdot |\mathcal{E}^+_C(\mathcal{L}') \cap \mathcal{E}^+(\mathcal{L})|$ of the right-hand side. For an edge to contribute to the term it must be in both $\mathcal{E}^+(\mathcal{L})$ and $\mathcal{E}^+(\mathcal{L}')$ and at least one of its endpoints in cluster $C$. This means that every edge in $\mathcal{E}^+(\mathcal{L}) \cap \mathcal{E}^+(\mathcal{L}')$ contributes exactly $\sum_{C:|\{u,v\} \cap C| \geq 1} z_C = 1 + x_{uv}$. Thus, we get:

$$\sum_C z_C \cdot |\mathcal{E}^+_C(\mathcal{L}') \cap \mathcal{E}^+(\mathcal{L})| = \sum_{uv \in \mathcal{E}^+(\mathcal{L}) \cap \mathcal{E}^+(\mathcal{L}')} (1 + x_{uv}) = |\mathcal{E}^+(\mathcal{L}) \cap \mathcal{E}^+(\mathcal{L}')| + \sum_{uv \in \mathcal{E}^+(\mathcal{L}) \cap \mathcal{E}^+(\mathcal{L}')} x_{uv}$$

By substituting these rewritten terms in the inequality, we get the following:

$$\text{cost}(\mathcal{L}') \leq 2\text{LP} - \text{LP}^- - \sum_{uv \in \mathcal{E}^+(\mathcal{L}') \cup \mathcal{E}^-(\mathcal{L}')} x_{uv} - |\mathcal{E}^+(\mathcal{L}) \cap \mathcal{E}^+(\mathcal{L}')| + 2 \sum_{uv \in \mathcal{E}^+(\mathcal{L})} x_{uv} - \sum_{uv \in \mathcal{E}^+(\mathcal{L}) \cap \mathcal{E}^+(\mathcal{L}')} x_{uv}$$

$$\leq 2\text{LP} - \text{LP}^- - \sum_{uv \in \mathcal{E}^+(\mathcal{L}') \cup \mathcal{E}^-(\mathcal{L}')} x_{uv} - |\mathcal{E}^+(\mathcal{L}) \cap \mathcal{E}^+(\mathcal{L}')| + 2 \sum_{uv \in \mathcal{E}^+(\mathcal{L})} x_{uv}$$

$$\leq 2\text{LP} - |\mathcal{E}^-(\mathcal{L}')| - \sum_{uv \in \mathcal{E}^+(\mathcal{L}')} x_{uv} - |\mathcal{E}^+(\mathcal{L}) \cap \mathcal{E}^+(\mathcal{L}')| + 2 \sum_{uv \in \mathcal{E}^+(\mathcal{L})} x_{uv}$$

For the second inequality we used the fact $\sum_{uv \in \mathcal{E}^+(\mathcal{L}) \cap \mathcal{E}^+(\mathcal{L}')} x_{uv} \geq 0$ and for the third inequality the fact $\text{LP}^- + \sum_{uv \in E^-(\mathcal{L}')} x_{uv} \geq \sum_{uv \in E^-(\mathcal{L}')} (1 - x_{uv} + x_{uv})$. Finally, if $\mathcal{L}'$ is not $(2 - \delta)$-approximate, then:

$$|\mathcal{E}^-(\mathcal{L}')| + \sum_{uv \in \mathcal{E}^+(\mathcal{L}')} x_{uv} + |\mathcal{E}^+(\mathcal{L}') \cap \mathcal{E}^+(\mathcal{L})| < \delta \text{LP} + 2 \sum_{uv \in \mathcal{E}^+(\mathcal{L})} x_{uv}$$

which is the desired statement. $\qquad\square$

