# OpenReview forum: "Towards Better-than-2 Approximation for Constrained Correlation Clustering"
_ICML.cc/2025/Conference — ICML 2025 spotlightposter_

### Official Review · Reviewer_h3bn · 2025-03-07

**Overall Recommendation:** 3

**Summary:**

In this paper, the authors study Constrained Correlation Clustering, introduced by van Zuylen and Williamson (2009). The problem is a generalization of the well-known Correlation Clustering, where we are additionally given the set of friendly pairs and the set of hostile pairs and the goal is to find an optimal clustering (in terms of Correlation Clustering) satisfying the additional constraint that every friendly pair is placed within a cluster and every hostile pair is placed across clusters. In the context of machine learning, this problem can be seen as a semi-supervised variant of Correlation Clustering. For the problem, the state-of-the-art polynomial-time approximation ratio is 3, which was also given by van Zuylen and Williamson (2009). In the present paper, the authors propose a polynomial-time better-than-2 (< 1.94) approximation algorithm. To this end, the authors combine two techniques that have recently been developed to achieve better-than-2 approximations for Correlation Clustering: one is the cluster LP (Cao et al., 2024) and the other is a local search algorithm based on an optimal clustering (Cohen-Addad et al., 2024).

## update after rebuttal
My initial recommendation was conditional due to an issue in the proof. The authors have now rectified this in their rebuttal, and I maintain my original recommendation.

**Claims And Evidence:**

Generally speaking, the claims are supported by clear and convincing evidence. In particular, the intuitive explanations provided throughout the paper make it easy to follow.

**Essential References Not Discussed:**

No essential references missing.

**Experimental Designs Or Analyses:**

No experiments performed.

**Methods And Evaluation Criteria:**

The proposed algorithm makes sense.

**Other Comments Or Suggestions:**

Comments:
- Regarding Constrained Correlation Clustering, unfamiliar readers could think that handling the set of friendly pairs is trivial (as they could think that the pairs can be simply aggregated into super elements). It would be better to point out that the aggregation leads to a weighted instance outside the problem.
- Considering that Constrained Correlation Clustering is a generalization of Correlation Clustering, we can see that the present paper also simplifies the analysis of some better-than-2 approximations for Correlation Clustering. This point could be better highlighted.
- Section 4 could be organized better using subsections etc.
- What the "fractional clustering" refers to should be explained.
- Lemmas 4.2 and 4.13 are trivial and do not need the proofs. Alternatively, Corollary 4.14 is not so straightforward and the proof would be helpful.
- On Page 5, we do not need to mention the range of the cost to see the polynomial time complexity of Algorithm 1.
- The reason why we are required to use the fractional clustering, as stated on Page 6, is not quite clear. Is it to simplify the analysis?
- The first sentence of the last paragraph of the proof of Lemma 4.16 is not clear.
- It might be unusual not to have a concluding section.

Typos:
- Page 2: Wirt -> Wirth
- Page 2: an $(1+\epsilon)$ -> a $(1+\epsilon)$
- Page 3: unweighted, undirected -> unweighted, undirected, complete
- Page 5: Some $E^+$ and $E^-$ should be $\mathcal{E}^+$ and $\mathcal{E}^-$, respectively.
- Lemma 4.10 is not exactly the same as Lemma 6 of Cao et al. (2024) due to the generalization.

**Other Strengths And Weaknesses:**

- I enjoyed reading the paper and was a bit surprised that a better-than-2 approximation for (Constrained) Correlation Clustering can be achieved through such a simple and reader-friendly analysis.
- Although it is acknowledged that the paper improves the state-of-the-art result of the SODA 2009 paper, the contribution might be not quite significant in the machine learning community. Indeed, the proposed algorithm is, unfortunately, not practical due to solving the variant of the cluster LP and the authors do not test the algorithm empirically.

**Questions For Authors:**

Please see the above comments.

**Relation To Broader Scientific Literature:**

Correlation Clustering is one of the most actively-studied clustering problems in machine learning.

**Theoretical Claims:**

I have checked the correctness of all contents in the main body. My recommendation, Weak Accept, is conditional, based on the following concern about the correctness. Let's look at the last paragraph of the proof of Lemma 4.4 (starting with "We similarly transform the right-hand side..."). The authors state that a plus edge in $\mathcal{E}^+(\mathcal{L})$ contributes an amount of $z_C$ to the sum if either $u$ or $v$ is contained in $C$. However, in my understanding, it is wrong. Indeed, a plus edge in $\mathcal{E}^+(\mathcal{L})$ contributes an amount of $z_C$ to the sum if both $u,v$ are contained in $C$. Note that the right-hand side computes the cost decremented by $C$ rather than that incremented by $C$. This leads to the contribution $1-x_{uv}$ rather than $1+x_{uv}$. Similarly, for a minus edge, it seems we have the contribution $1-x_{uv}$ rather than $1+x_{uv}$. Based on the above, it is not clear if we can still have the simplification stated in Lemma 4.4.

---

> ### Author Rebuttal · Authors · 2025-03-31
>
> We thank the reviewer for their thorough review and comments.
>
> First we reply to their two main concerns, that is, the practicality of our algorithm and Lemma 4.4.
>
> ---
>
> Regarding the practicality of our algorithm, as the reviewer noted, the bottleneck is in solving the Constrained Cluster LP. We are happy to share that after our submission, a paper titled “Solving the Correlation Cluster LP in Nearly Linear Time” was published at STOC 2025. This makes us optimistic that our overall algorithm can be practical.
>
> It is worth noting that to feasibly use this particular approach, one should compromise for worse theoretical guarantees. Furthermore, the paper is 57 pages long, so it would require a whole new work to study how to adapt it to the constrained setting.
>
> We view our algorithm as a flexible framework to which one can incorporate any solution to the Constrained Cluster LP (whether it is a variation of the STOC paper or some totally new algorithm), and get a solution for Constrained Correlation Clustering. Under this light, future work can focus on implementing practical solvers for the Constrained Cluster LP (even with weaker guarantees, as long as they are efficient) and get solutions for Constrained Correlation Clustering.
>
> ---
>
> Regarding Lemma 4.4, thank you very much for spotting the issue. There is indeed a mistake in the proof the way it is written in the submission. Specifically, we should have used a stronger inequality resulting from the local optimality of the clustering L. However, the following is a simple fix.
>
> We say that an edge is crucial if it has at least one endpoint in C. Since the cost of L is at most the cost of $L_C$, it must be the case that the number of crucial edges unsatisfied by $L_C$ is more than the number of crucial edges unsatisfied by L. We redefine $E_C^+(L)$ to be the set of crucial edges in $E^+(L)$, and $E_C^-(L)$ to be the set of crucial edges in $E^-(L)$.
>
> The proof, as was written in the submission, holds once the aforementioned changes are made.
>
> ---
>
> We also briefly reply to some of the other comments of the reviewer. For the ones we do not give a reply, we agree with the reviewer and will incorporate the suggestions on the paper.
>
> * "The reason why we are required to use the fractional clustering, as stated on Page 6, is not quite clear. Is it to simplify the analysis?":
> Indeed, we will make this more clear. This is not just to simplify the analysis. It is because the guarantees for our two clusterings are with respect to the fractional optimal (instead of the actual optimal, as was done in the paper that introduced the local search approach for correlation clustering [Combinatorial Correlation Clustering; Cohen-Addad et al.; STOC 2024]).
>
> * "On Page 5, we do not need to mention the range of the cost to see the polynomial time complexity of Algorithm 1.":
> Our understanding is that it takes polynomial time to improve the cost by at least 1, and as the maximum cost is polynomial, this improvement cannot happen many times. However this would not be the case if the maximum cost is exponential and every time we only improve by 1.
>
> * "What the "fractional clustering" refers to should be explained.":
> We will make this more clear; with "fractional clustering" we refer to a solution of the Constrained Cluster LP.
>
> * "The first sentence of the last paragraph of the proof of Lemma 4.16 is not clear."
> Indeed, we will rephrase that to "By maximality of S we have $Q_{XY'Z} \le S$, and therefore we can find an injective function $f: Q_{XY'Z} \rightarrow S$". If $\{u,f(u)\}$ is a plus edge [...]" so that we are more precise.
>
> * Conclusion section: Thanks for the suggestion. We will add a concluding section summarizing our contributions and discussing the research directions left open by our work.

---

> > ### Comment · Reviewer_h3bn · 2025-04-02
> >
> > Thank you for addressing my comments.
> >
> > I'm not entirely convinced that the algorithm becomes practical with the STOC'25 result, because: (i) While the cluster LP can theoretically be solved in linear (or even sublinear) time, this does not necessarily translate to strong practical performance. (ii) As the authors noted, it is non-trivial whether the fast algorithm applies to the constrained cluster LP. That said, I agree that referencing the STOC'25 paper helps highlight the potential practicality of the algorithm.
> >
> > I also believe the presentation of the results could be improved to better emphasize their significance to the ML community. One way to do this is by more strongly motivating Constrained Correlation Clustering in an ML context. Currently, it appears mainly as a semi-supervised variant of Correlation Clustering, but providing concrete application scenarios could strengthen the impact of the paper.
> >
> > Regarding the proof of Lemma 4.4, I'm glad to hear that the authors have corrected it.

---

> > > ### Author Response · Authors · 2025-04-04
> > >
> > > We agree with the reviewer and shall definitely refer to the STOC’25 paper in order to highlight the potential practicality aspect. As suggested by the reviewer, we shall also address the challenges associated with it.
> > >
> > > We shall also follow the suggestion to include a discussion on motivating the problem from the ML-perspective in the final version of the paper.
> > > In particular, there are specific use-cases in the ML literature that have used constrained correlation clustering in the way that we define it. As an example, constrained correlation clustering with must-link and cannot-link constraints has been used in clustering news articles about the same event across different languages [IJCAI 2007; Correlation Clustering for Crosslingual Link Detection; Van Gael, Zhu]. The hard constraints here were introduced in order to ensure that news articles about different events from the same language do not end up being in the same cluster.
> > > Finally, we would like to point out the broader applicability of must-link and cannot-link constraints, as evidenced by the works from the ML community incorporating these constraints, in general, for other clustering objectives.
> > >
> > > We once again thank the reviewer for their thorough comments.

---

### Official Review · Reviewer_Rox7 · 2025-03-08

**Overall Recommendation:** 5

**Summary:**

This paper proves a better-than-2 approximation for constrained correlation clustering (correlation clustering where certain "friendly" pairs are required to be in the same cluster and other "hostile" pairs are required to be separated). The approach combines two recent techniques for standard correlation clustering that led to better-than-2 approximations for the unconstrained case: rounding the cluster LP relaxation and the local search method. In more detail, this paper approximately solves the cluster LP relaxation (with new constraints for friendly and hostile pairs), runs local search, then runs another local search with a new type of penalty to get a qualitatively different clustering, and then rounds the LP relaxation to get another clustering. If the first two clusterings are no better than 2 approximations, a new procedure is used to "mix" them with the rounded LP solution to get another clustering that is a better-than-2 approx.

## Update post rebuttal

Thanks for the responses. I continue to have a very high view of the paper.

**Claims And Evidence:**

None of the claims are problematic, the paper provides proofs for all aspects of the paper.

**Essential References Not Discussed:**

None

**Experimental Designs Or Analyses:**

There are no experimental results.

**Methods And Evaluation Criteria:**

The methodology is sound and is based on a combination of existing techniques for getting better than two approximations for standard correlation clustering.

**Other Comments Or Suggestions:**

On the right column at the top of page 5, I think you are missing a "z_C" in one of your summations; you have \sum_{c : v \in C} = 1 instead of \sum_{c: v \in C} z_C = 1.

In the statement and proof for Corollary 4.5, you used E several places instead of \mathcal{E}

**Other Strengths And Weaknesses:**

This paper is overall very strong and fills a large gap in the literature. There has been a lot of work on unconstrained correlation clustering recently, with very little work focused on understanding how these recent advances can apply to other variants of correlation clustering. The paper outlines prior work and provides important background very clearly, and then provides a detailed proof for better-than-2 approximations for constrained correlation clustering. This builds on prior techniques for unconstrained correlation clustering, but the theoretical contribution is still very non-trivial. It needs to combine two previous techniques from the literature, and includes several other new tricks and results in order to make everything work for the constrained case. I found the manuscript very easy to read and the proof (while detailed and non trivial) is laid out very carefully and logically for the reader.

**Questions For Authors:**

On page 7 you mention that creating a new clustering that is 30\delta-competitive is a "contradiction" for small enough \delta. However, when I read this I had not forgotten that you previously set \delta to be 2/31, meaning that you get a slightly better than 2-competitive result, and unless I'm missing something there's no contradiction here. My understanding is that the idea is just that if your first 2 clusterings are not $2-\delta$ competitive, you then can construct a 3rd clustering via "mixing" that is a better than 2 approx. My question is: am I missing something or is this really not a proof by contradiction but just rather a proof that at least one of three clusterings is guaranteed to get you the desired result? I wonder if it'd be cleaner and more direct to not try to argue there is some sort of contradiction here. This is of course very minor.

This entire approach also applies when F = H = \emptyset, right? So can we not also view this as a new approach for getting a better-than-2 approximation for unconstrained correlation clustering that mixes a couple prior approaches? This would not be groundbreaking given the other existing better approximation factors, but could there be a benefit of your approach even just for unconstrained correlation clustering (e.g., in terms of simplicity of arguments?).

**Relation To Broader Scientific Literature:**

There has been a flurry of recent work on better-than-2 approximations for standard correlation clustering (which the paper has reviewed in detail), but very little for constrained correlation clustering. This paper shows how these recent methodologies (with some extra work) can be made to work for constrained correlation clustering.

**Theoretical Claims:**

I checked the first couple proofs in more detail (finding no problems), and skimmed more quickly through the rest of the proofs. I did not encounter any issues. The logical layout of the Lemmas and corollaries is very clear and does a good job guiding the reader.

---

> ### Author Rebuttal · Authors · 2025-03-31
>
> We thank the reviewer for their suggestions, we will apply them all in the final version.
>
> Regarding the proof-by-contradiction: Indeed, this claim of ours is inaccurate; this is how it was treated in the original local search paper [Combinatorial Correlation Clustering; Cohen-Addad et al.; STOC 2024], because the clustering obtained from "mixing" was only part of the analysis (as it requires knowledge of the optimal). As the reviewer correctly understood, in our case we have access to a (fractional) optimal, and therefore also have access to the clustering obtained by "mixing". Therefore, there is no need for proof-by-contradiction. We shall rectify that in the final version of the paper.
>
> We also thank the reviewer for their suggestion to highlight that our work conceptually simplifies better-than-2 approximations of (unconstrained) Correlation Clustering.

---

> > ### Comment · Reviewer_Rox7 · 2025-04-01
> >
> > Thanks for the reply.

---

### Official Review · Reviewer_W4nh · 2025-03-13

**Overall Recommendation:** 4

**Summary:**

The main contribution of this paper is the development of a 1.94-approximation algorithm for the constrained correlation clustering problem. In this context, the input is a graph consisting of edges labeled as {+1, -1}. The objective of correlation clustering is to find a partition (clustering) of the nodes that minimizes the total number of negative edges inside clusters plus positive edges between clusters. A natural extension of this problem is imposing constraint that dictates whether certain selected edges must belong to the same cluster or to different clusters.

For vanilla correlation clustering, there are two primary techniques LP and LS that achieve good approximation. Both struggle to be directed applied to the constrained version. However, the authors point out that the fractional solution is sufficient to help find a satisfactory integer solution. Roughly speaking, they initially form a collection of clusters by selecting all subsets with a positive indicator from the LP solution, and then compute a clustering L by running local search. Next, they compute L’ by running local search with penalty based on L. Finally, they implement the pivot algorithm, which “mixes” L and L’ and outputs a clustering P (with high prob). At least one solution out of {L, L’, P} achieves a 2-2/31 approximation. The authors demonstrate this by assuming that both L and L’ are inadequate, leading to the conclusion that P would be a sufficiently good solution.

**Claims And Evidence:**

All claims are supported by proofs.

**Essential References Not Discussed:**

I have not found so far.

**Experimental Designs Or Analyses:**

There is no experiments in this paper

**Methods And Evaluation Criteria:**

This is a totally theoretical paper without experiments.

**Other Comments Or Suggestions:**

typos

- Line 214, a feasible clustering $C$ should be $\mathcal{C}$
- In corollary 4.5, some $\mathcal{E}$ are misspelled as $E$

**Other Strengths And Weaknesses:**

The contribution of this paper is straightforward. The study provides an innovative approximation ratio for the constrained correlation clustering problem. Moreover, the overall idea of the proof is concise and easy to follow. The algorithm is very simple to implement.

**Questions For Authors:**

More discussions on the hardness are needed.
1. As the constrained correlation clustering is APX-Hard, are there any studies or results that discuss the lower bound of its approximation ratio?
2. Correlation clustering w/o constraints are APX-Hard. Are there more nuanced differences in computing complexity between the two? Maybe constrained correlation clustering is much harder.

Although responses to these questions will not affect my score, I believe it is beneficial to include this content in the paper for a more comprehensive discussion.

**Relation To Broader Scientific Literature:**

the method the author proposed has the potential to handle more constrained problems.

**Theoretical Claims:**

Yes. I checked all proofs except those in the supplement.

---

> ### Author Rebuttal · Authors · 2025-03-31
>
> We thank the reviewer for their suggestions, we will apply them all in the final version.
>
> In particular, regarding APX-Hardness of Correlation Clustering:
> - It was shown that Correlation Clustering is APX-Hard, but without an explicit constant, in [Clustering with qualitative information; Charikar, Guruswami, Wirth; FOCS 2003].
> - An explicit constant 24/23 > 1.043 was given in [Understanding the Cluster LP for Correlation Clustering; Cao et al.; STOC 2024], under the assumption that $P \ne BPP$. In the same paper they proved a 4/3 integrality gap for the cluster LP.
>
> We are not aware of any studies or results that discuss lower bounds on the approximation ratio of Constrained Correlation Clustering. In fact, when we first started this work, we were considering whether 2 is an actual lower bound, but we did not have any insight on how to leverage the hard constraints in order to improve the lower bound.
>
> It is worth mentioning here that Constrained Correlation Clustering is not the only problem generalizing Correlation Clustering for which the only known lower bound is the APX Hardness of (unconstrained) Correlation Clustering. Ultrametric Violation Distance and $L_1$ Best-Fit Ultrametrics (from the tree-reconstruction world) also generalize Correlation Clustering, but their hierarchical nature has not been successfully leveraged to provide any better lower bound.
>
> Once again, we thank the reviewer and will incorporate the above discussion in the final version of our paper.

---

> > ### Comment · Reviewer_W4nh · 2025-04-04
> >
> > Thanks for your clarification.

---

### Official Review · Reviewer_hg9m · 2025-03-13

**Overall Recommendation:** 5

**Summary:**

The authors consider the classic Correlation Clustering problem which, given a complete graph with edges labeled either + or -, the goal is to find a partition of the vertices so as to minimize the number of + edges across parts plus the number of - edges within parts. The has received a lot of attention since its introduction in the early 2000s.

The authors study the hard-constrained version of Correlation Clustering which goes as follows. In addition to the graph, some pairs of vertices may be labeled as hard positive or hard negative. The solution then must not separate any hard positive pair and not join any hard negative pair.

The problem is harder than the standard Correlation Clustering for which a 1.43 approximation is known and for which a 2.5-approximation has been known for 20 years. Only a 3-approximation was known for the hard-constrained version of the problem.

The authors combine two state-of-the-art methods to obtain this results: the Cluster Linear Program (LP) introduced at FOCS'23 and further analyzed at STOC'24, and the local search techniques presented at STOC'24. It is remarkable that these two techniques could be (1) extended to handle the hard-constrained version of the problem (albeit at the expense of worse approximation) and (2) combined to obtain better approximation bounds. The idea of using the clusters output by the Cluster LP as input clusters for local search is neat and a clever way to handle the hard constraints.

**Claims And Evidence:**

Yes

**Essential References Not Discussed:**

None.

**Experimental Designs Or Analyses:**

No experiments.

**Methods And Evaluation Criteria:**

Yes

**Other Comments Or Suggestions:**

I would suggest to remove "competitive" and use "approximation" in, e.g., 2-competitive. Usually competitive refers to the online algorithm setting, and it is a bit confusing to have both competitive and approximation at the same time in the paper.

**Other Strengths And Weaknesses:**

I like the paper. It cleverly combines state-of-the-art techniques to solve a more general problem.

**Questions For Authors:**

No specific question at this stage.

**Relation To Broader Scientific Literature:**

Nothing missing.

**Theoretical Claims:**

Yes, I went over the proofs. I haven't checked the proofs in the appendix in details though.

---

> ### Author Rebuttal · Authors · 2025-03-31
>
> We thank the reviewer for the suggestion. We inherited the "competitive" terminology from the local search paper [Combinatorial Correlation Clustering; Cohen-Addad et al.; STOC 2024], but we understand now that mixing the two can be confusing. We will stick to "approximate".

---

### Decision · Program_Chairs · 2025-05-01

**Decision:**

Accept (spotlight poster)

**Comment:**

Correlation Clustering is a fundamental “clustering under imperfect information” problem, several variants of which have been studied for more than two decades especially from an approximation-algorithms viewpoint. In the basic problem, we are given a complete graph G on n vertices with edges labeled either + or -, and one example objective is to efficiently find a partition of the vertices that minimizes the sum of the (number of + edges across parts) and the (number of - edges within parts).

This paper studies a natural version of the problem with hard constraints of the following form: some pairs of vertices may be labeled as “hard positive” or “hard negative”---we should not separate any hard-positive pair and must not keep any hard-negative pair in the same cluster. This problem is harder than standard Correlation Clustering.

The paper combines two state-of-the-art methods: the “Cluster Linear Program” (LP) and local-search techniques presented very recently in major theoretical CS conferences in the last two years. Expert reviewers found it remarkable that these two techniques could be extended to handle the hard-constrained version of the problem and combined to obtain better approximation bounds. The idea of using the clusters output by the Cluster LP as input clusters for local search was viewed as a particularly innovative way to handle the hard constraints.